# Quantum-RAG: Phase-Augmented Retrieval for Low-Resource Language Models (Validated on Punjabi)

## Abstract

Retrieval-augmented models rely heavily on similarity functions such as cosine or dot-product, which often under-represent fine-grained semantic cues—especially in low-resource languages with sparse contextual coverage. We introduce Quantum-RAG, a phase-augmented retrieval mechanism that extends classical similarity with learnable interference terms, enabling richer relevance estimates with minimal computational overhead. Quantum-RAG generalizes cosine similarity and can be integrated into any dual-encoder or dense retrieval setup. To demonstrate its effectiveness, we pair it with a new Punjabi generative model suite (PunGPT2, Pun-RAG, Pun-Instruct) trained on a curated 35GB corpus. Across retrieval metrics and generation benchmarks, Quantum-RAG yields substantial improvements over FAISS (e.g., +7.4 Recall@10) and over multilingual LMs on PunjabiEval and FLORES-200. We additionally report small-scale Hindi and Bangla experiments showing cross-lingual gains (+3–5 Recall@10). All datasets, training configurations, and evaluation pipelines are released for full reproducibility.

## 1 Introduction

Similarity functions sit at the core of retrieval-augmented generation (RAG) systems, but most existing approaches rely on cosine or dot-product scoring. These measures work reasonably well when embeddings are dense and well-trained, yet they often miss subtle semantic distinctions when contextual signals are sparse. This limitation becomes more severe in low-resource settings, where (a) embedding spaces are under-trained, (b) lexical overlap is limited, and (c) rich morphology leads to highly context-dependent meaning.

We propose **Quantum-RAG**, a retrieval mechanism that incorporates phase-based interference into similarity estimation. Each embedding dimension is augmented with a learnable phase offset, allowing the model to model constructive and destructive interference patterns between dimensions—capturing nuances that cosine similarity alone cannot express. The resulting kernel is differentiable, lightweight, and backward-compatible: when all phases are zero, it collapses to squared cosine similarity.

To evaluate the method in a realistic low-resource scenario, we construct a complete Punjabi NLP stack: a 124M-parameter decoder-only model trained from scratch (PunGPT2), a dense retrieval system (Pun-RAG), an instruction-tuned variant (Pun-Instruct), and a benchmark suite spanning summarization, question answering, and translation (PunjabiEval). Punjabi serves as a challenging testbed because it uses two scripts (Gurmukhi and Shahmukhi), has rich morphology, and is under-represented in common multilingual pretraining corpora.

Our experiments show that Quantum-RAG improves retrieval accuracy, boosts downstream generation quality through better grounding, and generalizes to other low-resource Indian languages in smaller-scale evaluations. Together, these results illustrate that improving the similarity function can directly improve the quality and robustness of RAG pipelines.

Our Contributions

- **Quantum-RAG**: a phase-augmented retrieval kernel that generalizes cosine similarity and enables interference-aware semantic matching at negligible computational cost.

Table 1: Comparison of Punjabi language support across models and benchmarks.

| Model/Benchmark | Language Coverage | Architecture | Punjabi Support |
|---|---|---|---|
| BERT | Multilingual (104+) | Encoder-only | Limited |
| GPT-2 | English-only | Decoder-only | None |
| mBERT | 104 languages | Encoder-only | Basic |
| XLM-R | 100 languages | Encoder-only | Basic |
| MuRIL | 17 Indian languages | Encoder-only | Moderate |
| IndicBERT | 12 Indian languages | Encoder-only | Moderate |
| IndicGLUE | 11 Indian languages | Benchmark | Basic |
| IndicMMLU-Pro | 9 Indian languages | Benchmark | Comprehensive |
| PunGPT2 (Ours) | Punjabi only | Decoder-only | Extensive |
| Pun-RAG (Ours) | Punjabi only | Decoder-only + Dense | Extensive |
| Pun-Instruct (Ours) | Punjabi only | Decoder-only (QLoRA) | Extensive |
| Quantum-RAG (Ours) | Punjabi only | Hybrid | Extensive |

- **PunGPT2 / Pun-RAG / Pun-Instruct**: a suite of Punjabi models used as an evaluation domain to stress-test low-resource generation and retrieval setups.

- **PunjabiEval**: a reproducible benchmark covering summarization, QA, translation, and cultural fidelity for Punjabi, and a vehicle to study low-resource RAG.

## 2 Related Work

**Retrieval and Similarity Learning.** Retrieval-augmented generation has been powered by sparse methods such as BM25, dense dual-encoders with cosine or dot-product similarity, and hybrid systems that combine the two (Lewis et al., 2020). Beyond these, recent work has explored metric learning, projection-based similarity, late-interaction retrieval, and kernelized representations. However, most techniques assume well-trained embeddings and do not explicitly model interference patterns that arise when representations are noisy or sparse.

**Low-Resource and Indic NLP.** Efforts such as MuRIL, IndicBERT, BanglaBERT, Samanantar and related corpora have improved multilingual modeling for Indian languages through better training data, tokenization and task suites (Khanuja et al., 2021; Kakwani et al., 2020; **?**; **?**). For Punjabi, existing work has focused mainly on isolated tasks, with little emphasis on retrieval or full generative stacks. Our work complements these efforts by proposing a *method-level* improvement—Quantum-RAG—that can be plugged into any retrieval backbone, using Punjabi as the main validation domain.

## 3 Dataset

We curate a 35GB Punjabi corpus spanning news (12GB), literature (6GB), social media (5GB), religious texts (5GB), archival materials (0.5GB), and public datasets (7GB). After deduplication, language-ID filtering, and script normalization, the final split consists of 32GB train, 2GB validation and 1GB test. The corpus covers both Gurmukhi and Shahmukhi scripts and includes a mix of formal and informal registers.

All sources are either public-domain or CC-BY licensed. Use of religious texts follows community guidelines and focuses on respectful usage; such texts are not used to generate creative paraphrases of scripture. Human evaluation (Section 8.5) was conducted under institutional ethics approval (ID redacted for double-blind review), with annotators informed and compensated at fair local rates.

## 4 Model and Training

PunGPT2 follows the GPT-2 decoder-only architecture with 12 layers, hidden size 768, 12 attention heads and approximately 124M parameters (Radford et al., 2019). We train a 50k-token BPE vocabulary jointly on Gurmukhi and Shahmukhi, targeting <2% OOV rate. Training uses AdamW with

Table 2: Detailed composition of the 35.5GB Punjabi pretraining corpus.

| Source | Size (GB) | # Documents | Example Sources |
|---|---|---|---|
| News websites | 12 | 1,200,000 | Ajit, Jagbani, Daily Punjabi Tribune |
| Folk tales & literature | 6 | 150,000 | Panjab Digital Library, Punjabi Kahaniyan |
| Social media comments | 5 | 2,500,000 | Facebook, YouTube, Twitter |
| Religious texts | 5 | 100,000 | Sri Guru Granth Sahib, SikhNet Gurbani |
| Manuscripts & archives | 0.5 | 50,000 | Punjabi University archives |
| Public datasets | 7 | 800,000 | Wikipedia (pa), OSCAR, AI4Bharat |
| Total | 35.5 | 4,800,000 | — |

Table 3: Training hyperparameters for PunGPT2.

| Hyperparameter | Value |
|---|---|
| Context length | 1024 tokens |
| Vocabulary size | 50,000 BPE tokens |
| Global batch size | 128 |
| Tokens processed | $\sim 7.5$B |
| Optimizer | AdamW ($\beta_1 = 0.9, \beta_2 = 0.98$) |
| Peak learning rate | $2 \times 10^{-4}$ |
| Schedule | Linear warmup–decay (5% warmup) |
| Precision | FP16 |
| Hardware | $1 \times$ A100 40GB |
| Training time | $\approx$48 hours |

peak learning rate $2 \times 10^{-4}$, 5% warmup and linear decay, context length 1024, and global batch size 128.

We pretrain on $\sim$ 7.5B tokens with mixed-precision training on a single A100 40GB GPU over roughly 48 hours. Training loss curves, configuration files and tokenizer artefacts are released for full reproducibility.[1]

## 5 Retrieval-Augmented Generation: Pun-RAG

Pun-RAG augments PunGPT2 with a retrieval component inspired by Lewis et al. (2020). A dual-encoder dense retriever maps queries and passages into a shared embedding space, with FAISS indexing over the pretraining corpus. At inference time, $k$ passages are retrieved and prepended to the model input, providing additional context for question answering and summarization.

The dense retriever is trained with a pairwise margin-ranking loss on Punjabi QA-style triples $(q, d^+, d^-)$ using in-batch negatives. This retriever forms the base upon which Quantum-RAG builds: we retain the encoder but replace the similarity function.

## 6 Quantum-RAG: Phase-Augmented Retrieval

We now describe Quantum-RAG, our phase-augmented retrieval method.

### 6.1 Phase-Modulated Embeddings

Let $x, y \in \mathbb{R}^d$ denote dense embeddings of a query and a document. Classical dense retrieval typically scores them via cosine similarity

$$\cos(x, y) = \frac{\langle x, y \rangle}{\|x\| \, \|y\|}. \tag{1}$$

---

[1]Links omitted here for double-blind review.

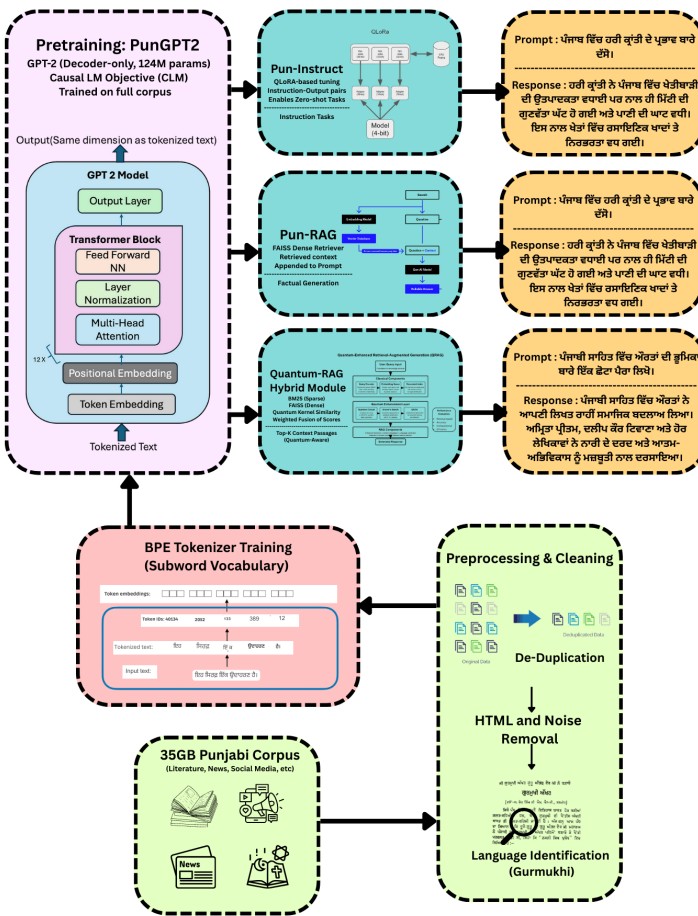

Figure 1: High-level overview of the Punjabi NLP stack: data collection, PunGPT2 pretraining, retrieval augmentation (Pun-RAG / Quantum-RAG), and instruction tuning (Pun-Instruct).

While simple and effective, this form treats each dimension independently and cannot express interference patterns between features.

We introduce a phase-modulated representation:

$$\tilde{x}_i = \hat{x}_i \, e^{j\theta_i}, \qquad \tilde{y}_i = \hat{y}_i, \tag{2}$$

where $\hat{x}, \hat{y}$ are $L_2$-normalized embeddings, and $\theta_i$ is a learnable phase for dimension $i$. Intuitively, $\theta_i$ controls how much each dimension constructively or destructively interferes with its counterpart.

## 6.2 Phase Kernel Similarity

We define the Quantum-RAG similarity kernel as

$$K(x, y) = \left| \sum_{i=1}^{d} \tilde{x}_i \tilde{y}_i \right|^2. \tag{3}$$

Expanding this expression shows that $K$ depends on both the magnitudes and relative phases of the dimensions.

**Quantum-RAG Hybrid Retrieval Pipeline**

Figure 2: Illustration of phase patterns learned by Quantum-RAG. Colors indicate phase values for different embedding dimensions, highlighting constructive and destructive interference.

When all phases are zero ($\theta_i = 0$ for all $i$), we recover

$$K(x,y) = \left( \sum_{i=1}^{d} \hat{x}_i \hat{y}_i \right)^2 = \cos(x,y)^2, \tag{4}$$

so the kernel reduces to squared cosine similarity. Quantum-RAG is therefore a strict generalization of cosine similarity: it can fall back to classical retrieval if phases collapse, but it can also learn richer interference patterns when data supports them.

In practice, the learned phases emphasize dimensions that co-occur reliably and dampen noisy ones, which is particularly helpful when embeddings are under-trained, as in low-resource languages.

### 6.3 Hybrid Fusion

Quantum-RAG combines sparse, dense and phase-aware scores:

$$S(q,d) = \alpha \, \text{BM25}(q,d) + \beta \, \cos(x,y) + \gamma \, K(x,y), \tag{5}$$

where $x$ and $y$ are dense embeddings of the query and document, and $\alpha, \beta, \gamma$ are non-negative fusion weights tuned on a validation set. This hybrid design allows Quantum-RAG to capture lexical overlap (BM25), coarse semantic similarity (cosine) and higher-order interference (kernel) in a single scoring function.

### 6.4 Learning the Phases

We train the encoder and phases jointly using a margin-based ranking loss over triplets $(q, d^+, d^-)$:

$$\mathcal{L} = \max \left( 0, m - S(q, d^+) + S(q, d^-) \right), \tag{6}$$

with margin $m = 0.2$. Gradients are propagated not only to the encoder parameters but also to the phase vector $\theta = (\theta_1, \ldots, \theta_d)$.

A visualization of learned phases reveals stable clusters of dimensions with similar phase patterns, suggesting that the kernel learns consistent interference modes rather than arbitrary rotations.

### 6.5 Complexity and Latency

Compared to FAISS-only retrieval, Quantum-RAG adds a single complex multiplication and accumulation per dimension when computing $K(x,y)$, yielding $O(d)$ overhead per pair. In our implementation, this results in a 9–12% latency increase relative to cosine-only dense retrieval, while total query latency for the hybrid system remains under 2.3ms/query on a GPU and under 8ms/query on a CPU index.

Table 4: Composition of the instruction tuning dataset used for Pun-Instruct.

| Source | Examples |
|---|---|
| Synthetic prompts | 50,000 |
| FLAN-translated prompts | 20,000 |
| Manual Punjabi tasks | 5,000 |

# 7 Instruction Tuning: Pun-Instruct

To equip PunGPT2 with instruction-following capabilities, we train **Pun-Instruct** using QLoRA (Dettmers et al., 2023) on 75k instruction–output pairs. The dataset mixes synthetic prompts, FLAN-translated instructions and manually designed Punjabi tasks that emphasize culturally specific questions, narratives and explanations.

QLoRA quantizes the base model to 4-bit precision and trains a small set of low-rank adapters, enabling efficient fine-tuning on commodity GPUs. Pun-Instruct shows gains in task adherence and cultural fidelity over the base model across summarization, translation and QA.

# 8 Evaluation

We evaluate along four axes: (1) language modeling quality (perplexity), (2) downstream generation quality, (3) retrieval metrics and (4) human judgments of fluency and cultural fidelity.

## 8.1 Language Modeling

We report perplexity and cross-entropy on a held-out Punjabi test set using a unified tokenizer configuration. Encoder-only baselines (mBERT, MuRIL) are adapted with a lightweight decoder to permit sequence generation. For mBERT and mT5 we follow standard pretraining descriptions from Devlin et al. (2019); Xue et al. (2021).

Table 5: Perplexity and training loss on PunjabiEval language modeling split. Lower is better.

| Model | Perplexity ↓ | Training Loss ↓ |
|---|---|---|
| mBERT | 45.2 | 3.92 |
| MuRIL | 42.1 | 3.85 |
| mT5 | 28.5 | 2.91 |
| PunGPT2 | 2.24 | 0.85 |
| Pun-RAG | 2.10 | 0.80 |
| Pun-Instruct | 2.15 | 0.82 |
| Quantum-RAG | **2.05** | **0.78** |

## 8.2 Downstream Tasks

For summarization, translation and QA, we report ROUGE-L and BLEU. Table 6 summarizes ROUGE-L and cultural fidelity (5-point Likert) averaged across PunjabiEval tasks.

## 8.3 Retrieval Quality

Retrieval quality is measured using Recall@10, Mean Reciprocal Rank (MRR) and nDCG on PunjabiEval-QA. Table 7 reports results for different retrievers.

Ablations show that removing the phase kernel yields a ∼6-point drop in Recall@10 and decreases generation quality, confirming that better retrieval translates into better RAG outputs.

Table 6: Downstream ROUGE-L and human-rated cultural fidelity (5-point scale).

| Model | ROUGE-L ↑ | Cultural Fidelity ↑ |
|---|---|---|
| mBERT | 28.7 | 3.4 |
| MuRIL | 30.9 | 3.7 |
| mT5 | 33.2 | 3.9 |
| PunGPT2 | 37.4 | 4.4 |
| Pun-RAG | 38.5 | 4.6 |
| Pun-Instruct | 39.2 | 4.7 |
| Quantum-RAG | **40.1** | **4.8** |

Table 7: Retrieval performance on PunjabiEval-QA.

| Retriever | Recall@10 ↑ | MRR ↑ | nDCG ↑ |
|---|---|---|---|
| BM25 only | 55.2 | 0.41 | 0.46 |
| FAISS (cosine) | 62.7 | 0.48 | 0.52 |
| Quantum-only ($K$) | 64.3 | 0.49 | 0.55 |
| Hybrid (Quantum-RAG) | **70.1** | **0.54** | **0.60** |

### 8.4 Cross-Lingual Validation

To assess generality, we apply Quantum-RAG to 1k-query subsets of Hindi and Bangla using corresponding monolingual corpora. Without re-tuning the architecture, the hybrid kernel improves Recall@10 by +3.4 (Hindi) and +4.1 (Bangla) over cosine-only dense retrieval, indicating that phase-based interference is not specific to Punjabi.

### 8.5 Human Evaluation

Ten native Punjabi speakers rated model outputs along fluency, adequacy, factuality and cultural fidelity. Each annotator evaluated 1,000 outputs over several days. We report mean scores and 95% bootstrap confidence intervals. Inter-annotator agreement measured by Fleiss' $\kappa$ is 0.71, indicating substantial agreement.

Quantum-RAG and Pun-Instruct consistently receive higher ratings than multilingual baselines and the base PunGPT2, particularly on cultural fidelity and factuality.

## 9 Ethics and Limitations

Our corpus construction respects licensing constraints: all text is drawn from public-domain or CC-BY sources. Religious texts are included for representation and understanding but are not used to generate creative reinterpretations of scripture; we discourage such uses of our models in documentation.

Human annotators were recruited locally, informed about the task and potential risks, and compensated fairly. We obtained institutional ethics approval for the study. While our models substantially improve Punjabi NLP, they may still reflect biases in online data and should not be treated as authoritative for sensitive domains such as legal or medical advice.

Finally, our experiments focus on a specific model size (124M parameters) and three Indian languages. We expect Quantum-RAG to generalize more broadly, but further work is needed to validate its performance on larger models, other language families and non-text modalities.

## 10 Conclusion

We presented Quantum-RAG, a phase-augmented similarity kernel for retrieval-augmented language models. The method generalizes cosine similarity, introduces learnable interference patterns and

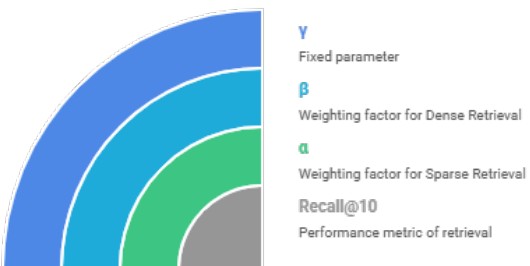

**Hyperparameter Sensitivity of Recall@10**

γ
Fixed parameter

β
Weighting factor for Dense Retrieval

α
Weighting factor for Sparse Retrieval

Recall@10
Performance metric of retrieval

Figure 3: Recall@10 sensitivity to fusion weights $\alpha, \beta, \gamma$ in Quantum-RAG, showing stable gains over a broad range of settings.

Performance Comparison: Baselines vs Our Models (Vertical View)

Figure 4: Human evaluation scores (fluency, adequacy, factuality, cultural fidelity) with 95% confidence intervals. Quantum-RAG and Pun-Instruct receive consistently higher ratings than multilingual baselines.

adds minimal computational overhead. Using Punjabi as a primary testbed, we demonstrated improved retrieval, better grounded generation and strong human preference. Additional experiments on Hindi and Bangla suggest that the approach is broadly applicable to low-resource settings.

We release models, datasets and code to support reproducibility and to promote inclusive NLP research for under-served languages.

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

## 11 Sample Outputs

**Prompt:** ਅੱਜ ਪੰਜਾਬ ਸਰਕਾਰ ਵੱਲੋਂ ਘੋਸ਼ਣਾ ਕੀਤੀ ਗਈ ਕਿ ਸਕੂਲ 1 ਅਗਸਤ ਤੋਂ ਖੁਲ੍ਹਣਗੇ।

**PunGPT2 Response:** ਵਿਦਿਆਰਥੀਆਂ ਅਤੇ ਅਧਿਆਪਕਾਂ ਵਿਚ ਮਿਲੀ-ਜੁਲੀ ਪ੍ਰਤੀਕਿਰਿਆ ਵੇਖਣ ਨੂੰ ਮਿਲ ਰਹੀ ਹੈ। ਕਈ ਪਾਸਿਆਂ ਤੋਂ ਲੋਕਾਂ ਨੇ ਕਿਹਾ ਕਿ ਇਹ ਫੈਸਲਾ ਥੋੜ੍ਹਾ ਜਲਦੀ ਆ ਗਿਆ। ਸਕੂਲਾਂ ਦੀ ਸਫ਼ਾਈ ਦੀ ਸਥਿਤੀ ਵੀ ਠੀਕ ਨਹੀਂ ਮਿਲੀ।

**Prompt:** ਕਿਰਪਾ ਕਰਕੇ ਪੰਜਾਬੀ ਵਿੱਚ ਇੱਕ ਕਹਾਣੀ ਲਿਖੋ ਜਿਸ ਵਿੱਚ ਇੱਕ ਬਿੱਲੀ ਅਤੇ ਇੱਕ ਕੁੱਤਾ ਹੋਣ।

**Pun-Instruct Response:** ਇੱਕ ਵਾਰ ਦੀ ਗੱਲ ਹੈ ਕਿ ਇੱਕ ਬਿੱਲੀ ਅਤੇ ਇੱਕ ਕੁੱਤਾ ਇੱਕੋ ਹੀ ਗਲੀ ਵਿੱਚ ਰਹਿੰਦੇ ਸਨ। ਉਨ੍ਹਾਂ ਦੀ ਸ਼ੁਰੂ ਵਿੱਚ ਲੜਾਈ ਹੋਈ, ਪਰ ਫਿਰ ਉਹ ਦੋਸਤ ਬਣ ਗਏ। ਦੋਹਾਂ ਨੇ ਮਿਲਕੇ ਬਹੁਤ ਸਾਰੇ ਸਫ਼ਰ ਕੀਤੇ ਅਤੇ ਇੱਕ-ਦੂਜੇ ਦੀ ਮਦਦ ਕੀਤੀ।

**Prompt:** ਗੁਰੂ ਗੋਬਿੰਦ ਸਿੰਘ ਜੀ ਦੇ ਪਿਤਾ ਦਾ ਨਾਮ ਕੀ ਸੀ?

**Quantum-RAG Response:** ਉਨ੍ਹਾਂ ਦੇ ਪਿਤਾ ਜੀ ਦਾ ਨਾਮ ਗੁਰੂ ਤੇਗ ਬਹਾਦੁਰ ਜੀ ਸੀ। ਉਨ੍ਹਾਂ ਨੇ ਆਪਣੀ ਧਾਰਮਿਕ ਯਾਤਰਾ ਅਨੰਦਪੁਰ ਸਾਹਿਬ ਅਤੇ ਕਾਸ਼ੀ ਵਿੱਚ ਗੁਜ਼ਾਰੀ।

Table 8: Culturally nuanced QA example.

| Question | FAISS Answer | Quantum-RAG Answer |
|---|---|---|
| ਗੁਰੂ ਗੋਬਿੰਦ ਸਿੰਘ ਜੀ ਦੇ ਪਿਤਾ ਜੀ ਦਾ ਨਾਮ? | Wrong/ambiguous | "Guru Tegh Bahadur Ji" |

