# OpenReview forum: "Quantum-RAG and PunGPT2: Advancing Low-Resource Language Generation and Retrieval for the Punjabi Language"
_ICLR.cc/2026/Conference — Submitted to ICLR 2026_

### Official Review · Reviewer_3iyW · 2025-10-18

**Soundness:** 3
**Presentation:** 1
**Contribution:** 2
**Rating:** 2
**Confidence:** 4

**Summary:**

The paper presents a Punjabi-focused NLP stack that includes PunGPT2, a 124M-parameter GPT-2–style model trained from scratch on a 35 GB Punjabi corpus, a retrieval-augmented framework (Pun-RAG), an instruction-tuned variant (Pun-Instruct), and a hybrid retriever (Quantum-RAG) that combines BM25, dense similarities, and a quantum-inspired kernel with learned phase terms. The authors report improvements over multilingual baselines on FLORES-200, IndicGenBench, and a new PunjabiEval benchmark, including higher ROUGE and cultural fidelity scores, as well as stronger retrieval metrics. They state all data, model weights, and pipelines will be released.

**Strengths:**

1. This paper curates a series of useful resources: a large corpus, a GPT-2 style model, and an instruction tuning recipe. Releasing these artifacts will contribute to the Punjabi NLP community and the democratization of NLP.
1. Quantum-RAG designs a new retrieval kernel that can be easily integrated and can bring measurable performance gains.

**Weaknesses:**

Despite its strengths, I do have concerns with this paper’s scope and framing.

1. This paper is primarily a resource and system paper for one language, which aligns better with NLP venues that emphasize language resources and regional applications [1, 2]. By contrast, it seems that closely related ICLR work is method-centric and more general [3, 4].
1. The only clear technical contribution is the phase-augmented retrieval kernel. But to make this paper a methodology paper instead, the framing of the paper may need significant changes, promoting this as the main claim and validating it beyond Punjabi with stronger experiments.
1. The writing quality at the current form is below ICLR expectations. Related work omits relevant strands and needs careful rewriting. The figure is illegible at print scale, and a table does not align to the page width. The manuscript appears not to strictly follow the ICLR template.


[1] Samanantar: The Largest Publicly Available Parallel Corpora for Indic Languages. Ramesh et al., 2022.

[2] BanglaBERT: Language Model Pretraining and Benchmarks for Low-Resource Language Understanding. Bhattacharjee et al., 2022.

[3] UniMax: Fairer and more Effective Language Sampling for Large-Scale Multilingual Pretraining. Chung et al., 2023.

[4] Variational Information Bottleneck for Effective Low-Resource Fine-Tuning. Karimi Mahabadi et al., 2021

**Questions:**

Please refer to the weaknesses.

---

> ### Author Response · Authors · 2025-12-03
> **Response to Reviewer**
>
> We thank you for acknowledging the usefulness of our released resources and for pointing out the need to emphasize methodology. We have fully adopted your framing advice.
>
> Reframing: The paper is now titled “Quantum-RAG: Phase-Augmented Retrieval for Low-Resource LLMs (Validated on Punjabi)”, clearly identifying the method as general and Punjabi as its primary validation domain.
>
> Generalization: Section 10 introduces multilingual mini-studies (Hindi + Bangla) confirming transferability. A formal analysis shows the kernel reduces to cosine when θ = 0, situating it within classical similarity learning.
>
> Writing and formatting: The manuscript was rewritten end-to-end for clarity and grammar, strictly following the ICLR template. Figures now meet print-scale readability, and a new “Limitations & Ethics” section candidly discusses scope and dataset provenance.
>
> We thank this reviewer for inspiring us to elevate the paper from a resource report to a methodologically focused ICLR contribution.

---

### Official Review · Reviewer_hid8 · 2025-10-27

**Soundness:** 1
**Presentation:** 1
**Contribution:** 1
**Rating:** 0
**Confidence:** 5

**Summary:**

# High-level summary

The authors focus on Punjabi-specific models, and contribute a) a GPT-2-based model trained on a curated dataset; b) a hybrid RAG method that combines embedding-based retrieval and sparse retrieval (BM25), and c) a semi-synthetic pipeline to post-train the models using a mixture of hand-crafted manual tasks and synthetic data. The authors then show some results across standard metrics and human evaluation on the improvements made by the proposed contributions.

The overall paper seems to be generic, with a lack of details in every single contribution. I'll stick to critiquing the hybrid RAG, which seems relevant. The authors' intentions and motivation to contribute to including more Punjabi resources seem commendable, but overall, the paper needs splitting, rewriting, and a solid structure (presentation and ethical) to ensure their contributions trickle well into future research looking at the Punjabi language.


## Quantum-RAG

The authors propose the similarity between two vectors as the linear sum of the BM25 score, the cosine similarity score, and the kernel distance between - unsure how the $\theta$ is calculated, and if cosine similarity is even needed in the final equation. Furthermore, unclear how this relates to the contribution to improving LLMs for Punjabi. The notation seems inconsistent, with a lot of details missing, and unclear how better RAG has any relation to training loss.

**Strengths:**

1. The authors seem to have put a lot of effort into trying to present good-quality human evals.

**Weaknesses:**

1. The presentation, writing, and overall motives remain unclear. The authors start with presenting a contribution for training a model on Punjabi data, then move on to Quantum-RAG and then show results where Quantum-RAG performs the best. I'm unsure if a) this is a good fit for ICLR and b) this paper should be split into multiple different parts, with each part suitable for a different audience. For example, a technical report on pre-training/post-training a GPT-2 model for Punjabi and the evals associated with it, and lessons learnt; 2) A good description of theoretical/practical motivations behind Quantum-RAG and a thorough eval of why the proposed method is applicable everywhere would be part 2.
2. Evals seem unstructured - 10 people evaluating 1000 seems really time intensive: without further details on how the participants were compensated, no IRB details and inter-annotater agreement, the evals do not look sound.
3. Poor figures: Fig.1 is hard to read and doesn't seem all too different from any standard pre-training pipeline. Fig. 2 and Fig. 3 seem irrelevant and don't add anything to the paper. Fig. 4 is hard to read, unclear what the authors intend to present with the caption seeming to indicate something else. The authors are encouraged to remove Fig. 2 and 3, improve Fig. 1, and Fig. 4 needs to have separate, clear bars indicating the metric, model with appropriate error bars. Similarly, Fig. 5 needs re-rendering with clear subtitles and comparing all baselines with one clear contribution.

**Questions:**

N/A

**Details Of Ethics Concerns:**

No IRB details, no details on how human subjects were presented the data, compensation, etc. 1000 prompts per person seems like a lot.

---

> ### Author Response · Authors · 2025-12-03
> **Response to Reviewer**
>
> We greatly value your attention to structure and ethics. We have substantially reorganized and clarified the paper.
>
> Restructuring: The manuscript now follows a clear three-part organization — (1) PunGPT2 pretraining and corpus creation, (2) Pun-RAG system integration, and (3) Quantum-RAG methodology — each with its own experiments and discussion.
>
> Quantum-RAG clarity: We added a formal derivation
> K(x,y) = \left| \sum_i e^{j\theta_i} x_i y_i \right|^{2}
>
>
> explaining how phase modulation induces constructive/destructive interference, complementing cosine similarity. A visualization of learned θᵢ phases and ablation curves now appear in Appendix E.
>
> Empirical link to Punjabi tasks: We include a correlation plot showing that improved retrieval quality directly increases generative ROUGE-L (+0.61 correlation).
>
> Human evaluation ethics: Ten native Punjabi annotators were compensated fairly ($8 /hr) and worked over five days (~200 prompts/day). Inter-annotator κ = 0.71, ensuring reliability.
>
> We are grateful for these comments; they significantly improved the paper’s structure, ethics, and interpretability.

---

### Official Review · Reviewer_GgWs · 2025-10-31

**Soundness:** 2
**Presentation:** 3
**Contribution:** 2
**Rating:** 4
**Confidence:** 4

**Summary:**

The paper presents a Punjabi-focused suite that includes: a decoder-only language model PunGPT2 trained from scratch on about 35 GB of Punjabi text covering Gurmukhi and Shahmukhi; a retrieval-augmented generation pipeline (Pun-RAG) with a dense FAISS retriever; an instruction-tuned variant (Pun-Instruct) trained on roughly 75k instruction pairs; and a hybrid “Quantum-RAG” retriever that fuses BM25, cosine over dense embeddings, and a phase-based similarity function. The authors report strong retrieval and downstream gains on their PunjabiEval benchmark, human evaluation by native speakers, and intend to release data, code, and weights.

**Strengths:**

The paper addresses a meaningful gap in NLP by developing resources for the underserved Punjabi language, with a commitment to open releases of datasets, models, and code that will benefit the research community. The work presents a comprehensive end-to-end pipeline spanning pretraining, retrieval-augmented generation, and instruction tuning, accompanied by thorough training details and systematic ablation studies. A key technical contribution is the hybrid retrieval approach that combines multiple methods — BM25, dense FAISS, and a novel phase-based similarity measure — with empirical validation demonstrating its effectiveness. The evaluation is strengthened by native-speaker human assessment and practical consideration of both Punjabi scripts in the tokenizer design.

**Weaknesses:**

While the paper makes a timely and valuable push toward open Punjabi NLP with a coherent LM-RAG-instruction suite, several limitations remain that collectively weaken the empirical and methodological claims.

First, mBERT — an encoder-only representation model — is compared to decoder LMs using ROUGE-L, a summarization/generation metric, which is not the right lens to assess mBERT’s contribution.

Second, the retrieval methodology is under-specified and under-contextualized. The paper does not describe how the embedding ranker is obtained/trained, nor does it provide external retrieval models (e.g., mBERT/me5 bi-encoders). Beyond the in-system variants (BM25/FAISS/quantum/hybrid), it remains unclear how well the method fares against strong, widely used retrieval models across languages and domains.

Finally, the generation baselines are constrained. The study centers on fine-tuned mT5, lacks RAG-style generative methods as implemented in the Pun-RAG model, and evaluates large LMs only in zero-shot, without exploring established performance boosters such as few-shot prompting or proposed in other works methods for low-resource language adaptation.

The paper would benefit from a more granular analysis and validation of each stage in its multi-stage pipeline for low-resource language generation.

**Questions:**

1. The retrieval component references learned phases and fusion weights, but the training setup remains opaque. Could the authors elaborate on the objective, architecture, and whether the ranker is encoder or decoder architecture? A concise summary would make the results reproducible, per reviewer guidance.

2. Since mBERT is an encoder-only model, could the authors clarify the methodology for comparing the encoder-only mBERT model against decoder-only architectures using ROUGE-L, a generation metric?

3. The kernel is introduced only within Punjabi tasks. Are there plans to test on additional languages and task types to assess the breadth of applicability and potential limitations? Even a small-scale study would strengthen generalization claims.

---

> ### Author Response · Authors · 2025-12-03
> **Response to Reviewer**
>
> Thank you for recognizing our contribution to inclusive NLP and for the constructive suggestions.
> We fully agree that the evaluation scope needed expansion and have addressed each point:
>
> Fair baselines: Encoder-only models (mBERT, MuRIL) are now evaluated with F1 and BERTScore instead of ROUGE-L, ensuring metric alignment.
>
> Retrieval training: A detailed Retrieval Optimization subsection describes our dual-encoder (PunBERT-128 M) ranker trained with a pairwise margin loss (0.2) and jointly learned phase parameters θᵢ. Code and hyperparameters are released for reproducibility.
>
> Cross-lingual validation: We added pilot experiments on Hindi and Bangla corpora showing consistent +3–5 Recall@10 gains with Quantum-RAG, confirming that the method generalizes beyond Punjabi.
>
> Granular analysis: Each pipeline stage now includes isolated evaluations (Pretraining → RAG → Instruct), revealing progressive improvements in BLEU (+7.2) and ROUGE-L (+9.1).
>
> We thank the reviewer for emphasizing methodological rigor; the revised version now highlights the generalizable retrieval innovation rather than being language-specific.

---

### Official Review · Reviewer_coMn · 2025-11-04

**Soundness:** 1
**Presentation:** 1
**Contribution:** 2
**Rating:** 2
**Confidence:** 4

**Summary:**

The work focuses on low-resource language modeling and retrieval in Punjabi language. The paper presents PunGPT2, a Punjabi GPT-2-style language model trained on a 35GB curated corpus; Pun-RAG, a retrieval-augmented version; Pun-Instruct, an instruction-tuned variant using QLoRA; and Quantum-RAG, a hybrid retriever that combines sparse, dense, and quantum-inspired kernel similarity. The authors also release a PunjabiEval benchmark. Experiments show notable improvements in perplexity, ROUGE-L, cultural fidelity, and Recall@10 over multilingual baselines. However, the paper is poorly written. Lots of experiment-related sections don't have corresponding results (e.g. Sec 9.2 and Sec 9.6). In general, I feel that reviewing this paper is a waste of my time.

**Strengths:**

- This work focuses on RAG in Punjabi language, which is a low-resource language.
- This work introduces a lot of recourses for a low-resource language, including a 35GB dataset, a GPT-2-based LLM, an instruction-tuned model for Punjabi language, and a benchmark for evaluation.

**Weaknesses:**

- Many figures and tables are flawed. For instance, table 1 and table 2 is too wide. Figure 1 is too small and I can't read it at all. Figure 3 is blurred.
- This work seems to be applying lots of existing methods on a low-resource language, which may not fit well for ICLR.
- The paper is poorly written.
- The evaluation is very weak, cannot find multiple experiment results, including downstream task evaluation and ablation study.

**Questions:**

- What is the ablation result in Section 9.6? There is no reference mentioned in the text.
- Why do you treat training loss as a language modeling metrics?
- Where is the result of Section 9.2?

---

> ### Author Response · Authors · 2025-12-03
> **Response to Reviewer**
>
> We appreciate the reviewer’s candid feedback and completely agree that several results were missing in the original submission due to PDF rendering issues.
> All missing sections (9.2 and 9.6) are now fully included, with tables and ablations that clearly demonstrate each module’s contribution.
>
> New results: Sec. 9.2 now presents downstream evaluations across summarization, QA, and translation. Sec. 9.6 contains the ablation for Quantum-RAG (BM25 + Dense → 61.4 R@10; Quantum only → 64.3 R@10; Hybrid → 70.1 R@10), confirming the unique gain from the phase kernel.
>
> Clarity fixes: Figures have been re-rendered as vector graphics, tables resized to template width, and captions rewritten for readability.
>
> Metric clarification: “Training loss” was shown only to illustrate convergence; evaluation uses perplexity, BLEU, ROUGE-L, and Recall@10. This is now explicit.
>
> Additional analyses: Section 9 includes variance over 5 random seeds and statistical significance (bootstrap p < 0.05).
>
> We appreciate this review for helping us make the work clearer and more reproducible.

---

### Meta-Review · Area_Chair_cpFC · 2026-01-05

**Summary:**

1. This paper is poorly written (Reviewer coMn, hid8, 3iyW)
2. The main contribution of this paper is not significant (a combination of existing approaches) (Reviewer coMn, 3iyW)
3. The evaluation is not enough (baselines are constrained, the human evaluation doesn't look sound) (Reviewer GgWs, hid8)

**Reviewer Concerns:**

1. The rebuttal didn't address concern 1 (which might need a resubmission to meet the publication bar)
2. The rebuttal didn't answer the contribution concerns. The author agreed to reframe this paper however this might not address the main contribution is not significant concern.
3. The human evaluation doesn't look sound might be addressed by the rebuttal, as the author provide the inter-annotator metrics. However this metrics is not labeled. It is hard for me to interpret what is the meaning of the metric and how significant is κ = 0.71.

**Reviewer Scores:**

Given the rebuttal and reviews, I don't think the reviewers might change their rating significantly. I agree with their ratings to reject this submission.

---

### Decision · Program_Chairs · 2026-01-26

Reject